# Targeting Macrophages in Glioblastoma: Current Therapies and Future Directions

**DOI:** 10.3390/cancers17162687

**Published:** 2025-08-18

**Authors:** Giovanni Pennisi, Federico Valeri, Benedetta Burattini, Placido Bruzzaniti, Carmelo Lucio Sturiale, Andrea Talacchi, Fabio Papacci, Alessandro Olivi, Giuseppe Maria Della Pepa

**Affiliations:** 1Department of Neurosurgery, Fondazione Policlinico Universitario A. Gemelli IRCCS, 00168 Rome, Italy; federicovaleri97@gmail.com (F.V.); benedetta.burattini@gmail.com (B.B.); carmelo.sturiale@policlinicogemelli.it (C.L.S.); fabio.papacci@policlinicogemelli.it (F.P.); alessandro.olivi@policlinicogemelli.it (A.O.); giuseppemaria.dellapepa@policlinicogemelli.it (G.M.D.P.); 2Department of Neurosurgery, F. Spaziani Hospital, 03100 Frosinone, Italy; 3Department of Neurosurgery, ASST Santi Paolo e Carlo, 20153 Milano, Italy; placntz@gmail.com; 4Department of Neurosurgery, San Giovanni-Addolorata Hospital, 00168 Rome, Italy; atalacchi@hsangiovanni.roma.it

**Keywords:** tumor microenvironment (TME), glioblastoma (GBM), tumor-associated macrophages (TAMs), TAM target therapy

## Abstract

Glioblastoma (GBM) is a highly aggressive brain tumor with an immunosuppressive tumor microenvironment (TME) that drives resistance to therapy. Tumor-associated macrophages (TAMs), including microglia and bone marrow-derived macrophages, promote tumor growth and immune evasion, especially in their M2-like form. This systematic review analyzed 30 studies on TAM-targeted therapies in GBM. These approaches include blocking TAM recruitment, boosting their phagocytic activity, and reprogramming TAMs. Combination strategies with immune checkpoint inhibitors, nanoparticles, and oncolytic viruses showed enhanced anti-tumor effects in preclinical models. However, clinical success is limited by TAM plasticity, poor drug delivery across the blood–brain barrier, and immunosuppressive treatments. Future therapies should focus on patient-specific approaches using detailed TME profiling and advanced delivery systems to improve treatment response and overcome current limitations.

## 1. Introduction

Glioblastoma (GBM) is the most aggressive type of primary brain tumor, accounting for almost 48.6% of all malignant central nervous system (CNS) tumors.

The incidence is slightly higher in males than females, with most cases diagnosed in individuals over 60. Despite the best available treatments, the overall survival remains limited to around 12 to 15 months [1,2].

The current standard of care involves maximal safe surgical resection, followed by fractionated radiotherapy with concurrent daily temozolomide (TMZ). Additionally, bevacizumab and tumor treating fields (TTFs) are approved components in selected patients. Despite this aggressive multimodal approach, the infiltrative nature of GBM means that recurrence is nearly inevitable [2,3].

The tumor microenvironment (TME) of GBM is a highly dynamic ecosystem consisting of malignant cells, stromal components, immune cells, and extracellular matrix elements, all contributing to tumor progression and therapy resistance. A key feature of the TME is its ability to promote immune evasion by suppressing anti-tumor immune responses and creating conditions that help a tumor survive [1,4,5]. Within this TME, tumor-associated macrophages (TAMs), which include both brain-resident microglia and bone marrow-derived macrophages, play a crucial role in modulating immune activity, promoting angiogenesis, and facilitating tumor invasion.

Macrophages are classified into two major phenotypes: M1 (classically activated, pro-inflammatory) and M2 (alternatively activated, anti-inflammatory) [4]. M1 macrophages are typically induced by IFN-γ and bacterial components such as LPS (Lipopolysaccharide), and are characterized by high production of reactive oxygen species (ROS), IL-12, and TNF-α—molecules associated with anti-tumor activity [6,7]. In contrast, M2 macrophages are further subclassified into M2a, M2b, and M2c phenotypes, each with distinct stimuli and functional profiles. M2a macrophages are induced by IL-4 and IL-13 and contribute to tissue remodeling and wound healing. M2b macrophages respond to immune complexes and LPS and possess immunoregulatory properties, while M2c macrophages are induced by IL-10 and glucocorticoids and play a role in immunosuppression and matrix deposition. Despite this classification, TAMs in GBM often exhibit overlapping features, reflecting their high degree of plasticity and context-dependent functional states.

Although TAMs have been extensively studied in GBM, their role in lower-grade gliomas (LGG) remains less well-defined. Emerging evidence suggests that LGGs harbor fewer infiltrating macrophages and exhibit a less immunosuppressive TME. The reduced presence of M2-polarized TAMs in LGGs may explain, in part, their slower progression and better prognosis compared to GBM. Nonetheless, TAMs in LGG still contribute to local immunomodulation and may represent early therapeutic targets to prevent malignant transformation.

Given the limited success of conventional immunotherapies in treating GBM, there is a growing interest in targeting TAMs to modulate the TME and reinstate anti-tumor immunity. New strategies are being developed to eliminate TAMs, shift them toward a more pro-inflammatory M1 phenotype, or prevent their accumulation in the tumor environment. Additionally, combining TAM-targeted treatments with other immunotherapies, such as CAR T-cell therapy and oncolytic viruses, may help counteract the immune-suppressive nature of the GBM tumor environment.

This manuscript reviews the latest advancements in TAM-targeted therapies for GBM, discussing their mechanisms of action, preclinical and clinical efficacy, and potential challenges in their translation to clinical practice. By elucidating the dynamic interplay between TAMs and GBM cells, we aim to provide insights into how macrophage-targeted strategies could enhance the effectiveness of current and future GBM treatments.

## 2. Materials and Methods

This systematic literature review was conducted in accordance with the Preferred Reporting Items for Systematic Reviews and Meta-Analyses Protocols (PRISMA-P, see Figure 1) guidelines and has not been registered [8]. The literature search was performed using PubMed, Scopus, and Web of Science databases. The search strategy included the following keywords: “glioblastoma”, “tumor microenvironment”, “macrophage-targeted therapy”, “immune modulation”, and “tumor-associated macrophages”. The final search was conducted on 31 January 2025, and no restrictions on publication date were applied to ensure a comprehensive analysis. Two independent reviewers, G.P. and F.V., screened articles for eligibility. Any discordance was solved by consensus with a third senior author (G.M.D.P.). No restrictions on the date of publication were made. The following criteria were used to identify studies to be included in the review: original studies focusing on TAMs, their role in GBM progression, and therapeutic strategies targeting these cells. The exclusion criteria included the following: studies published in languages other than English, review studies, and meta-analyses. A systematic review of the references was conducted to identify additional records. After removing duplicates, titles and abstracts were screened, and full texts of relevant articles were reviewed. Discrepancies between reviewers were resolved through consensus with a senior author. A total of 2895 records were initially identified, and after screening, 31 full-text articles were included in the final analysis.

## 3. Results

A total of thirty articles were eligible for this review (see Figure 1). Although most molecules that target tumor-associated macrophages can have multiple effects on the complex balance of the tumor microenvironment, this review aims to provide a clear interpretative framework by categorizing TAM-targeting effects into three main functional domains: **(1) inhibition of TAM recruitment; (2) enhancement of TAM-mediated phagocytic activity; and (3) modulation of the macrophage equilibrium within the TME (Figure 2)**. Considering the complexity of this topic, and to enhance reader comprehension, we first introduce a section that clarifies the origin, features, and functions of TAMs before exploring the details of TAM-targeted therapies.

### 3.1. TAM Diversity in GBM: Origins, Markers, and Dynamics

TAMs in glioblastoma consist of bone marrow-derived macrophages (BMDMs) and brain-resident microglia, each playing distinct roles in tumor progression [6]. BMDMs originate from circulating monocytes and infiltrate the brain in response to tumor signals, which promotes immune suppression and enhances tumor invasion. In contrast, microglia, which come from embryonic yolk sac progenitors, stay in the brain throughout life and are more involved in local immune surveillance and tissue remodeling. While microglia regulate early immune responses, macrophages are more actively engaged in tumor growth and resistance to therapy. Traditionally, macrophages and microglia have been distinguished by CD45 levels, but this method is unreliable in GBM, where microglia can upregulate CD45 expression [6,9]. Advanced single-cell technologies have identified more accurate markers: macrophages express CCR2, CD45RA, and CD49D, while microglia are enriched in CX3CR1, SALL1, and P2RY12. Therefore, using multiple markers is essential for accurately distinguishing between these two cell populations. From a morphological perspective, microglia are larger cells with branched extensions that allow them to monitor the brain environment. In contrast, macrophages are more mobile, have fewer branches, and possess a greater ability to infiltrate tumors. Their distinct functions indicate that microglia play a role in maintaining brain integrity, while macrophages are involved in promoting tumor invasion and immune evasion [6].

The concept of macrophage plasticity in neuroimmunology has been extensively discussed [1,6,7,10], particularly concerning the traditional M1/M2 classification of macrophages. These cells were originally categorized into two main types: “classically activated” (M1) and “alternatively activated” (M2). M1 macrophages are known for their pro-inflammatory and tumor-fighting properties. They are stimulated by cytokines such as IFN-γ and TNF-α, which lead to the production of reactive oxygen species (ROS) as well as the pro-inflammatory cytokines IL-1β, IL-6, IL-12, and IL-23. These factors collectively contribute to an anti-tumor response. On the other hand, M2 macrophages are induced by cytokines like IL-4, IL-10, or IL-13. They secrete anti-inflammatory cytokines, including IL-10, TGF-β, CCL22, and CCL17, which support immune suppression and aid in tumor progression. The traditional dichotomy used to categorize macrophages has proven useful, but emerging evidence indicates that it oversimplifies the functional diversity of tumor-associated macrophages (TAMs) in glioblastoma (GBM) [4,9,10]. Firstly, the M1/M2 model is primarily based on in vitro studies. In contrast, in vivo analyses reveal that TAMs in GBM do not neatly fit into these categories. There is no clear correlation between the markers identified in vitro and those found in GBM tumors [9,10]. Secondly, defining TAMs based solely on select markers, such as CD163 and CD206 for M2 polarization, is insufficient because specific markers exclusive to each phenotype are lacking [10]. 

Furthermore, TAMs should be defined as a spectrum, with individual cells often co-expressing both pro-inflammatory and anti-inflammatory genes. Finally, TAM phenotypes are highly dynamic and can shift in response to the tumor microenvironment (TME) and therapeutic interventions. This complexity underscores that the M1/M2 framework alone is inadequate for fully understanding the functions of TAMs in GBM [6,10]. Understanding the diversity of tumor-associated macrophages (TAMs) is essential for developing targeted therapies.

### 3.2. Strategies for Inhibition of TAM Recruitment in GBM

A promising strategy in GBM research focuses on targeting the recruitment of tumor-associated macrophages by blocking the TAM infiltration into the tumor microenvironment (see Table 1). Various methods have been explored, ranging from blocking key interactions between chemokines and their receptors to inhibiting the upstream signaling pathways that drive TAM accumulation. Among these, the CCL2/CCR2 signaling axis has been extensively studied. Tumor cells release CCL2, a potent chemoattractant for CCR2-expressing myeloid cells, including tumor-associated macrophages (TAMs) and myeloid-derived suppressor cells (MDSCs). This process facilitates their migration into the tumor microenvironment (TME) [10,11].

The myeloid-derived suppressor cells (MDSCs) originate from the bone marrow like the bone marrow–derived macrophages (BMDMs); on the other hand, the MDSCs represent distinct myeloid cell populations with different stages of maturation and immunological roles. BMDMs are fully differentiated macrophages recruited into the glioma microenvironment, often assuming an M2-like phenotype that promotes tumor growth. In contrast, MDSCs are immature myeloid cells with potent immunosuppressive activity that inhibit T-cell responses and can differentiate into various cell types, including macrophages. MDSCs, particularly the monocytic subtype (M-MDSCs), are increasingly recognized as relevant contributors to the immunosuppressive milieu in GBM and share pathways with TAM recruitment. Therapeutic strategies targeting MDSCs, such as CCR2 inhibition, overlap with TAM-directed approaches [11,22].

Joseph A. Flores-Toro et al. proposed a CCR2 antagonist identified as CCX872. They investigated the potential of combining CCR2 inhibition with PD-1 blockade to improve treatment outcomes in gliomas resistant to anti-PD-1 therapy. Their findings revealed that the absence of CCR2 enhances the effectiveness of PD-1 blockade in KR158 glioma-bearing mice, leading to a previously unseen survival advantage. In their models, CCR2 deficiency was associated with a reduction in the presence of immunosuppressive myeloid-derived suppressor cells (MDSCs) within the tumor, while their numbers increased in the bone marrow. The CCR2 antagonist CCX872, when used alone, extended median survival, and when combined with anti-PD-1 therapy, it further prolonged both median and overall survival [11]. Hye Rim Cho et al. further investigated the CCL2/CCR2 pathway’s potential role. Their experiments with CCL2-expressing GBM cell lines (U87 MG and LN-18) showed that CCL2 promotes angiogenesis indirectly through macrophage attraction. When macrophages were treated with the CCL2 inhibitor mNOX-E36, they did not migrate toward tumor-conditioned media, confirming the necessity of CCL2 for their recruitment. In vivo, a rat model of GBM was employed to examine the therapeutic effects of combining mNOX-E36 with the anti-angiogenic drug bevacizumab. The findings revealed that tumors with elevated CCL2 levels were resistant to bevacizumab, a resistance alleviated by CCL2 inhibition [12]. Zhang et al. reported the role of Ginsenoside RK3 in modulating the PPARG (Peroxisome Proliferator-Activated Receptor-gamma)/CCL2 pathway within glioblastoma. RK3 downregulates PPARG expression in tumor cells, leading to a reduction in CCL2 secretion [13]. By inhibiting this pathway, RK3 decreases M2-type macrophage recruitment in the tumor microenvironment, promoting a more anti-tumor immune response.

Regarding other pathways involved the molecular adhesion and TAM recruitments, Lei Tian et al. developed a novel oncolytic herpes simplex virus type 1 (oHSV) designed to enhance the immune response against glioblastoma (GBM) [14]. This engineered virus, named OV-Cmab-CCL5, was modified to express a secretable single-chain fragment of cetuximab—an antibody targeting the epidermal growth factor receptor (EGFR)—fused to the chemokine CCL5 (chemokine C-C motif ligand 5). This strategy enabled the virus to selectively direct the production of CCL5 to EGFR-expressing tumor cells within the glioblastoma microenvironment. Once inside the tumor, OV-Cmab-CCL5 not only disrupted EGFR signaling, which is often upregulated in GBM, but also triggered a strong immune response. The virus plays a crucial role in modulating the recruitment and activation of immune cells, such as natural killer (NK) cells, macrophages, and T-cells. Consequently, treatment with OV-Cmab-CCL5 resulted in a significant reduction in tumor size and improved survival rates in mouse models of glioblastoma [14].

The programmed cell death-1 (PD-1) and programmed cell death ligand-1 (PD-L1) pathways are important immunological checkpoints that have been thoroughly researched. They have drawn considerable attention for their effectiveness in treating various aggressive tumors by negatively regulating T-cell-mediated immune responses. However, in glioblastoma (GBM), the immunosuppressive microenvironment limits the effectiveness of immune checkpoint inhibitors (ICIs) when used alone, leading to poor outcomes for patients [23]. Several studies have explored the use of anti-PD-L1 antibodies in combination therapies [15,16]. Notably, Tsubasa Miyazaki and colleagues investigated the effects of combining anti-PD-L1 antibodies with an M2 macrophage inhibitor called Eganelisib (referred to as IPI-549) on tumor growth. Their results demonstrated that the combination therapy of the anti-PD-L1 antibody and IPI-549 significantly inhibited tumor growth. Moreover, treatment with the anti-PD-L1 antibody markedly reduced the infiltration of CD163-positive macrophages (a marker of M2 macrophages) into the tumors. The combined therapy of the PD-L1 antibody and IPI-549 resulted in remarkable suppression of tumor growth [15].

Another important pathway involved the Hypoxia-Inducible Factor-1α (HIF-1α), a transcription factor that becomes highly activated under low oxygen conditions. HIF-1α regulates the expression of several genes involved in tumor adaptation to hypoxia, including stromal cell-derived factor 1 (SDF-1α). Research has shown that treatments like radiation therapy can increase tumor hypoxia, leading to elevated levels of both HIF-1α and SDF-1α, which in turn enhance the recruitment of macrophages to the tumors [18,24]. Similarly, anti-VEGF therapies, such as bevacizumab, have been observed to raise the expression of HIF-1α and SDF-1α, contributing to the tumor’s ability to evade treatment.

HIF-1α has also been implicated in the response of the tumor microenvironment (TME) to radiotherapy. Ionizing radiation, while causing direct DNA damage to tumor cells, also induces vascular dysfunction, resulting in hypoxia within the TME. This hypoxic state stabilizes hypoxia-inducible factor 1-alpha (HIF-1α), which promotes the polarization of macrophages toward an M2-like, pro-tumor phenotype. These M2-like macrophages are characterized by increased secretion of angiogenic factors (e.g., VEGF, TGF-β) and immunosuppressive cytokines (e.g., IL-10). Such polarization contributes to radioresistance by enhancing tumor revascularization, facilitating immune evasion, and supporting the repair of irradiated stromal tissue [25,26].

Olaptesed pegol (OLA-PEG) inhibits SDF-1α, preventing macrophage infiltration and enhancing anti-VEGF therapy’s effectiveness in an in vivo study [17]. Additionally, zoledronate (ZOL), delivered using specialized nanoparticles (ZOL@CNPs), not only induces apoptosis in temozolomide (TMZ)-resistant glioblastoma (GBM) cells but also encourages macrophages to adopt an anti-tumor M1 phenotype [18]. By inhibiting HIF-1α, ZOL@CNPs further combat the immunosuppressive and hypoxic microenvironment. Together, OLA-PEG and ZOL@CNPs disrupt macrophage-mediated tumor resistance, offering promising strategies for enhancing GBM treatment.

While the increase in macrophages following anti-VEGF therapy is widely accepted, the predominant TAM subpopulation remains uncertain [24]. Axitinib, a VEGFR tyrosine kinase inhibitor, alone or in combination with oncolytic herpes simplex virus, significantly elevated CD68+ macrophages in vivo [19]. However, CD68 is expressed not only by M1-like macrophages but also by other TAM subpopulations, including immunosuppressive ones, its use as a sole marker limits interpretation. Without additional markers, it remains unclear whether these macrophages contribute to tumor suppression or progression.

Plerixafor (AMD3100), a small molecule antagonist of CXCR4, has been widely studied in hematologic malignancies and is now being repurposed for glioblastoma. The CXCR4 receptor and its ligand CXCL12 (also known as SDF-1α) play critical roles in the recruitment of bone marrow–derived myeloid cells, including monocytes and endothelial progenitor cells, to the tumor microenvironment [24]. This recruitment contributes not only to immunosuppression but also to post-treatment vasculogenesis—an alternative to angiogenesis that becomes particularly relevant following anti-VEGF therapy or radiotherapy [27]. In preclinical models of GBM, Plerixafor effectively reduced the infiltration of CXCR4⁺ myeloid cells and enhanced radiosensitivity, suggesting a synergistic potential when combined with conventional treatments. Early-phase clinical studies have confirmed the safety and feasibility of combining Plerixafor with radiotherapy and have reported encouraging biological effects, including reduced TAM density and altered perfusion dynamics [21,27]. These findings underscore the therapeutic relevance of disrupting the CXCR4/CXCL12 axis in glioblastoma as part of a broader strategy to modulate the tumor microenvironment and overcome resistance to standard therapies.

Nanotechnology has been applied by Koula et al., who created doxorubicin-loaded cerium oxide nanoparticles coated with oleyl amine-linked cyclic RGDfK peptide (CeNP + Dox + RGD) to target both gliomas and their tumor microenvironment (TME) via integrin receptors [20]. Immunohistochemical analysis of brain tumor tissues revealed a significant modulation and blocking of TAM recruitment.

### 3.3. Enhancement of TAM-Mediated Phagocytic Activity

The ability of TAMs to phagocytose tumor cells is another key area of research. In this context, CD47, also known as integrin-associated protein, has long been recognized as a key mediator of anti-phagocytosis signaling (see Table 2). When CD47 binds to SIRPα on the surface of macrophages, it triggers a signaling cascade that inhibits phagocytosis [28,29]. Many authors described how blocking the CD47/SIRP-α pathway might be a target therapy against various cancers [30,31].

In glioblastomas, the CD47 pathway has been shown to significantly enhance M1-mediated phagocytosis of glioma cells, resulting in slowed tumor growth and prolonged survival times, with minimal to no side effects observed in both in vitro and in vivo models [29,32,34,39].

Combination therapy was explored by Von Roemeling et al., who described how combining temozolomide (“TMZ”) with anti-CD47 Ab showed increased anti-tumoral effects compared to monotherapy [32]. Gholamin et al. reported that the combination therapy with irradiation to effectively increases anti-CD47 Ab efficacy [29]. The same authors, in a previous study, targeted SIRP-α directly via a small molecule, Hu5F9-G4, reporting increased anti-tumor phagocytosis, with minimal activity against normal neural cells, both in in vitro and in patient-derived xenografts [33].

Being a promising pathway, researchers have also moved towards nanotechnologies and tissue engineering with the use of oncoviruses. Li et al. developed nanoparticles loaded with TMZ, anti-CD47 Ab, and a specific dye (“PA1094T”) which serves as a transporter for the nanoparticles under photon and acoustic waves, directly to the tumor cells. In a recurrent GBM model, the authors demonstrated how the combination of PA1094T and anti-CD47 Ab significantly enhanced cancer cells phagocytosis and remodeled the TME. Xu et al. developed an oncolytic herpes virus that expresses a full-length anti-human CD47 IgG1. This virus selectively replicates in glioma cells, prompting them to secrete anti-CD47 IgG into the tumor microenvironment while avoiding release into the systemic circulation. As a result, this approach has improved the survival rates of mice bearing gliomas [35].

Wang et al. developed a self-assembling paclitaxel (PTX) filament hydrogel that contains an aCD47, a hydrophilic macromolecular antibody [36]. This hydrogel can be directly applied into the surgical cavity, allowing for long-term and localized release of both therapeutic agents. This approach suppresses tumor recurrence and prolongs survival while minimizing distant side effects.

In addition to targeting the CD47/SIRPα axis, the pharmacological inhibition of BACE1 (β-site amyloid precursor protein-cleaving enzyme 1) using the inhibitor MK-8931 enhances TAM-mediated phagocytosis and hinders GBM progression in vivo [37].

Scolaro et al. demonstrated that extracellular UDP released by tumor cells promotes an immunosuppressive phenotype in tumor-associated macrophages (TAMs) via the P2Y_6_ receptor. Inhibition of this pathway—either by targeting P2Y_6_ in TAMs or cytidine deaminase (CDA) in tumor cells—restored cytotoxic T-cell infiltration and sensitized tumors to anti-PD-1 therapy [40]. Although the study focused on melanoma and colorectal cancer models, the findings are mechanistically relevant to GBM, where TAM-driven immune suppression similarly limits immunotherapeutic efficacy.

Zhan et al. explored a different approach by indirectly modulating the tumor microenvironment (TME) through alterations in the extracellular matrix (ECM) of the central nervous system (CNS) [38]. They focused on uridine diphosphate (UDP)-glucose 6-dehydrogenase (UGDH), the rate-limiting enzyme responsible for the biosynthesis of the primary glycosaminoglycans (GAG) in the CNS [41]. By using 4-methylumbelliferone (4-MU), a small molecule inhibitor of GAG synthesis, they increased the toxicity of cancer cells. 4-MU is already approved for human use and has demonstrated efficacy in various cancers [42]. In models of glioblastoma (GBM), it operates through multiple mechanisms, including enhanced phagocytosis by macrophages [38].

### 3.4. Modulation of the TAMs Equilibrium Within the TME

The modulation of TAMs involves targeted changes to their phenotypic and functional characteristics, with the goal of shifting from an immunosuppressive M2-like state to a pro-inflammatory M1-like phenotype. This approach, often called the reprogramming of TAMs, is particularly significant in glioblastoma (see Table 3).

The CSF-1/CSF1R axis (colony-stimulating factor-1 and colony-stimulating factor-1 receptor) is widely studied because it plays a crucial role in macrophage differentiation and survival [54].

Pyonteck et al. used a CSF-1R inhibition, called BLZ945, to therapeutically modulate TAMs in a murine model of proneural GBM [43]. This treatment significantly prolonged survival and induced regression of tumors. Moreover, BLZ945-mediated CSF-1R inhibition attenuated the intracranial growth of patient-derived glioma xenografts [43]. Unexpectedly, TAMs were not depleted following treatment. Instead, glioma-secreted factors such as granulocyte–macrophage colony-stimulating factor (GM-CSF) and interferon-gamma (IFN-γ) sustained TAM viability despite CSF-1R blockade. Surviving TAMs displayed a reduction in M2-associated gene expression, indicating a shift away from tumor-supportive phenotypes.

Justyna M. Przystal et al. proposed using a CSF1R inhibition antibody in single or combination therapy with a PD-1 antibody in a glioma mouse model and patient-derived micro-tumors (PDMs) [44]. Their findings indicated that blocking CSF1R alone could delay the onset of neurological symptoms, suggesting a potential benefit in slowing disease progression. However, the real breakthrough occurred when CSF1R inhibition was combined with PD-1 blockade. This dual therapy improved survival rates and resulted in long-term survivors, indicating a significant enhancement in the immune response against the tumor.

Furthermore, inhibiting CSFR signaling alters the balance between M2-like and M1-like TAMs, promoting the differentiation of monocytes into M1-like TAMs while suppressing the genetic and functional characteristics of M2-like TAMs [45].

Despite this promising result, a phase II study led by Nicholas Butowski involving the CSF1R/KIT inhibitor (PLX3397), an oral small molecule that penetrates the brain, showed that the drug was well-tolerated but had poor clinical outcomes [46]. 

Additionally, engineered molecular systems with immunostimulatory properties have been developed and studied as part of innovative therapeutic strategies for GBM. A study conducted by Xiaojun Wang and colleagues focused on extracellular vesicles derived from M1-like macrophages (M1EVs), which were engineered to deliver a combination of therapeutic agents [47]. These agents include the hydrophobic compounds CPPO and Ce6 embedded in the membrane, and the hydrophilic prodrug AQ4N contained within their core.

The resulting complex, known as CCA-M1EVs, was able to cross the blood–brain barrier and reprogram M2 macrophages into an anti-tumor M1 phenotype. Furthermore, the vesicles facilitated the conversion of hydrogen peroxide (H_2_O_2_) into reactive oxygen species (ROS) for chemiexcited photodynamic therapy while simultaneously worsening tumor hypoxia, which activated AQ4N into its cytotoxic form. This strategy led to significant tumor inhibition in both cell lines and patient-derived glioblastoma models, demonstrating promising translational potential for multimodal therapy in GBM treatment [47].

Another pathway explored for its immunostimulatory properties is the STING pathway (stimulator of interferon genes). Berger et al. evaluated the status of STING signaling in glioblastoma (GBM) tissues and investigated the immunomodulatory effects of the STING agonist ADU-S100 (synthetic cyclic dinucleotide -CDN) on the brain tumors microenvironment (TME) [48]. In murine models, the intracranial administration of ADU-S100 using biodegradable implants resulted in significant immunological changes in the TME. Specifically, activating STING led to extensive recruitment of innate immune cell populations to the tumor-bearing hemisphere, including inflammatory macrophages, neutrophils, and natural killer (NK) cells. These alterations were associated with increased survival rates and the induction of durable immune memory. Importantly, STING activation not only encouraged the infiltration of tumor-associated macrophages but also likely repolarized them toward a pro-inflammatory phenotype. This shift contributed to dismantling the immunosuppressive microenvironment often seen in GBM.

The mTOR (mechanistic target of rapamycin) pathway, which regulates oncogenic receptor tyrosine kinase (RTK) signaling, has been studied in glioblastoma (GBM) with EGFR mutations. Specifically, pharmacological inhibition of both mTORC1 and mTORC2 using agents like AZD8055, WYE-125132, MTI-31, and rapamycin resulted in a downregulation of tissue factor (TF), a pro-thrombotic molecule that is elevated in aggressive tumors and linked to poor prognosis. Notably, targeting this pathway led to significant changes in the tumor microenvironment in vivo, including a marked reduction in M2-like TAMs infiltration, indicated by a decreased CD206/F4/80 ratio [49].

Oncolytic viruses are emerging as promising immunotherapeutic tools for the treatment of GBM. The autonomous protoparvovirus H-1 (H-1PV) has been evaluated in the Phase I/IIa ParvOryx01 trial, where it demonstrated safety and the ability to cross the blood–brain barrier following intravenous administration [50]. H-1PV induced immunomodulatory effects within the tumor microenvironment, including increased infiltration of cytotoxic T-cells and activation of tumor-associated microglia/macrophages, as evidenced by the upregulation of cathepsin B, iNOS, and accumulation in CD40L-positive tumor regions [50,51].

Using the same treatment strategy, an oncolytic viral therapy derived from herpes simplex virus 2 (HSV-2) and adenovirus (DNX-2401) has been described in both in vivo and in vitro studies [52,53]. The authors demonstrate substantial anti-tumor activity and favorable tolerance in glioblastoma (GBM). The extraordinary efficacy of oncolytic viruses stems from their unique mechanisms: they selectively target the replication of tumor cells, induce powerful cytotoxic DNA damage stress, and enhance the activity of M1 macrophages by modulating and reprogramming the tumor microenvironment.

The combination effect of oncolytic viral therapy has been explored by Saha et al., who investigated the effects of combining a PD-L1 antibody with an anti-CTLA-4 treatment and an oncolytic herpes simplex virus in a mouse model of glioma. While this approach led to a weak extension of survival, on the other hand, authors reported significant immune changes within the tumor [16]. The treatment promoted an influx of macrophages, which showed a shift toward an M1-like, pro-inflammatory phenotype. 

## 4. Discussion

This systematic review examines therapeutic strategies targeting tumor-associated macrophages in glioblastoma. GBM is a type of cancer known for its highly immunosuppressive tumor microenvironment, which contributes to treatment resistance and disease recurrence [1]. Among TAMs, the M2-polarized phenotype is particularly significant in GBM progression, as it promotes angiogenesis, immune evasion, and tissue invasion. Therefore, targeting TAMs presents a promising opportunity to reprogram the tumor microenvironment (TME) and enhance clinical outcomes [1,6].

One major strategy focuses on inhibiting TAM recruitment, particularly through the CCL2/CCR2 chemokine axis. Preclinical models have shown that CCR2 antagonists such as CCX872 reduce the accumulation of immunosuppressive myeloid cells within tumors and enhance the efficacy of PD-1 checkpoint blockade, ultimately improving survival in glioma-bearing mice [11]. Similarly, mNOX-E36, a CCL2 inhibitor, impairs macrophage chemotaxis and re-sensitizes bevacizumab-resistant GBM models [12]. Inhibition of hypoxia-induced factors like HIF-1α and SDF-1α—upregulated during anti-VEGF therapy—also limits TAM infiltration and enhances tumor sensitivity to radiotherapy and anti-angiogenic treatment [24].

A second approach involves enhancing the phagocytic function of TAMs. Blockade of the CD47–SIRPα axis, a critical “don’t eat me” signal exploited by glioma cells, has demonstrated preclinical efficacy [30,31,33,34,39]. Antibodies such as Hu5F9-G4 promote macrophage-mediated engulfment of glioma cells, resulting in tumor regression and prolonged survival. When combined with standard treatments such as temozolomide or radiotherapy, anti-CD47 therapies have produced synergistic anti-tumor effects without damaging non-tumoral neural tissues [33]. Delivery methods, including nanoparticles and oncolytic viruses encoding anti-CD47 IgG1 have shown improved tumor specificity and reduced systemic exposure [35]. A third therapeutic avenue targets TAM polarization by reprogramming M2-like macrophages toward a pro-inflammatory M1-like phenotype. Inhibitors of the colony-stimulating factor 1 receptor (CSF1R), including BLZ945 and PLX3397, reduce M2-associated gene expression and improve survival in animal models [43]. These agents do not eliminate TAMs but rather shift their functional phenotype, indicating successful polarization. When CSF1R inhibition is combined with immune checkpoint blockade, particularly anti-PD-1 antibodies, murine models display enhanced immune activation and durable tumor responses [44]. Ex vivo data from patient-derived micro-tumors (PDMs) co-cultured with autologous tumor-infiltrating lymphocytes (TILs) corroborate these findings, with the combination of anti-CSF1R and anti-PD-1 promoting cytotoxic activity [44].

However, clinical translation remains challenging, and poor outcomes have been reported [46].

One key obstacle is the routine use of corticosteroids, such as dexamethasone, which are commonly administered postoperatively in GBM patients and are known to suppress T-cell activation, potentially undermining immunotherapy efficacy. Neoadjuvant checkpoint blockade—delivered before surgery and steroid use—has been proposed as a strategy to circumvent this limitation and potentiate immune priming [55,56].

Resistance mechanisms to TAM-targeted therapies are increasingly recognized [4,5,6,57].

One mechanism reported regarding the paracrine interaction between the tumor microenvironment (TME) and tumor cell’s receptor that activates the IGF-1R/PI3K pathway. These interactions maintain tumor-associated macrophage (TAM) viability and suppress therapeutic efficacy despite CSF1R inhibition [5]. This compensatory survival pathway operates independently of tumor-intrinsic genetic alterations and may be particularly relevant in GBMs harboring frequent PI3K pathway activation.

Accurate evaluation of preclinical and clinical studies critically depends on the potential bias of the methodologies used to assess the tumor microenvironment (TME). Despite encouraging preclinical data, the clinical relevance of TAM-targeted therapies remains uncertain. Several studies rely solely on CD68 expression to quantify macrophage infiltration; however, CD68 is not exclusive to M1-like macrophages and is also expressed by immunosuppressive M2 subsets [16]. Without simultaneous evaluation of additional markers (e.g., CD163 and CD206 for M2; CD80 and CD86 for M1), the biological significance of observed changes in macrophage populations remains unclear [4,6,9,10]. Moreover, the spatial and phenotypic heterogeneity of TAMs, along with the overlapping functions of resident microglia and bone marrow-derived macrophages, complicates therapeutic targeting. TAM plasticity further allows for dynamic adaptation to therapeutic pressure, making sustained modulation difficult to achieve [6,10].

Drug delivery is another significant barrier. Most biologic agents, including anti-PD-L1 monoclonal antibodies, are hindered by their inability to cross the blood–brain barrier (BBB) and the blood–tumor barrier (BTB), particularly when their molecular weight exceeds 400–600 kDa. This severely limits intra-tumoral bioavailability and likely contributes to the underwhelming performance of PD-L1-targeted therapies in GBM clinical trials despite high levels of PD-L1 expression in tumors [23]. Furthermore, reliable biomarkers to stratify patients or monitor TAM modulation in vivo are lacking. 

Moreover, studies have identified a gradient of activation involving microglia, macrophages, and other immune cells, which may play a critical role in therapeutic resistance and tumor recurrence [6,58,59]. Interestingly, such activation has also been observed in brain regions distant from the actively proliferating tumor core areas that do not show clear neoplastic infiltration or contrast enhancement on imaging [6]. These findings suggest that the inflammatory response associated with glioblastoma is widespread and not confined to the immediate tumor margin.

Although several TAM-targeting strategies have shown promise in preclinical models, translation into clinical efficacy remains inconsistent. For instance, CSF1R inhibitors such as BLZ945 demonstrated significant TAM depletion but failed to induce sustained tumor regression unless combined with radiotherapy or immune checkpoint blockade. Conversely, targeting the CCL2/CCR2 axis with agents like CCX872 enhances checkpoint efficacy but may not be sufficient alone. Strategies enhancing phagocytosis, like anti-CD47 antibodies, show strong immunostimulatory potential but may require careful dosing to avoid hematologic toxicity. Overall, TAM reprogramming strategies appear to offer the most flexibility, yet the heterogeneity of TAM subtypes and plasticity remains a key barrier to uniform response.

Looking forward, future efforts should prioritize patient-specific strategies grounded in comprehensive TME profiling. Rational combination therapies involving checkpoint inhibitors, metabolic agents, or PI3K/IGF-1R pathway inhibitors may overcome compensatory resistance. The development of advanced delivery systems capable of traversing the BBB and achieving sustained, localized immune modulation is essential. High-resolution spatial transcriptomics and intravital imaging technologies will play a critical role in mapping TAM heterogeneity and identifying actionable subpopulations. While TAMs represent a promising and multifaceted therapeutic target in GBM, realizing their full clinical potential will require integrated, multimodal approaches tailored to the complexity of glioblastoma biology.

### Limitations

This review specifically focused on therapeutic strategies targeting tumor-associated macrophages (TAMs) in glioblastoma. Given the complexity and breadth of the tumor microenvironment, we recognize that this targeted approach inevitably excludes certain overlapping and highly relevant biological processes, such as the effects of radiotherapy on TAM behavior, the effect on vasculogenesis and angiogenesis, and the role of myeloid cells in modulating vascular remodeling and immune response. Although mechanistically related, these aspects were beyond the scope of our inclusion criteria. Despite this limitation, our main goal was to provide an updated and focused overview of the most promising molecular strategies currently being studied to modulate TAM function. We believe this synthesis helps advance the understanding of a rapidly evolving and therapeutically important field.

## 5. Conclusions

Tumor-associated macrophages are an important but challenging target in glioblastoma therapy. Although many preclinical studies have shown that targeting TAMs can slow tumor growth and improve treatment response, results in clinical trials have been less promising. Several issues limit the success of these therapies, including the difficulty in correctly identifying TAM subtypes, their ability to change function in response to treatment, and the overlapping roles of different immune cells in the brain. The use of corticosteroids and the poor ability of many drugs to cross the blood–brain barrier also reduce treatment effectiveness. In addition, tumors may develop resistance through alternative survival pathways, such as IGF-1R/PI3K signaling.

To improve outcomes, future research should focus on personalized approaches that take into account the unique features of each patient’s tumor microenvironment. Combining TAM-targeted therapies with immune checkpoint inhibitors or drugs that block resistance pathways may enhance treatment responses. Better drug delivery systems that can reach brain tumors, along with new technologies like spatial transcriptomics and real-time imaging, will be critical for identifying and monitoring specific TAM populations. In summary, while TAMs remain a promising target, their clinical potential in GBM will only be fully realized through comprehensive and carefully tailored treatment strategies.

## Figures and Tables

**Figure 1 cancers-17-02687-f001:**
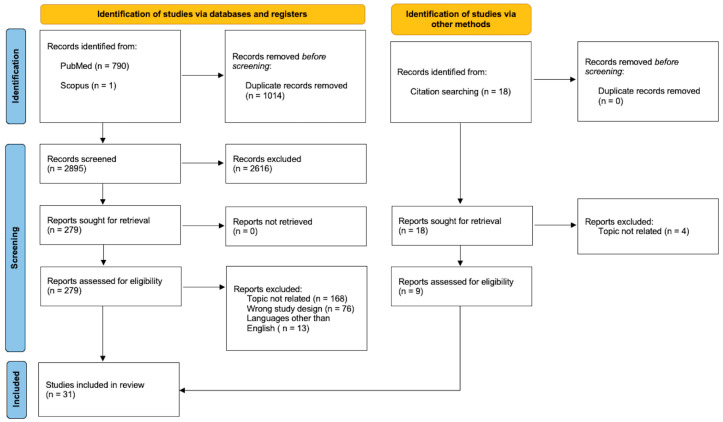
Prisma Flow Diagram [8].

**Figure 2 cancers-17-02687-f002:**
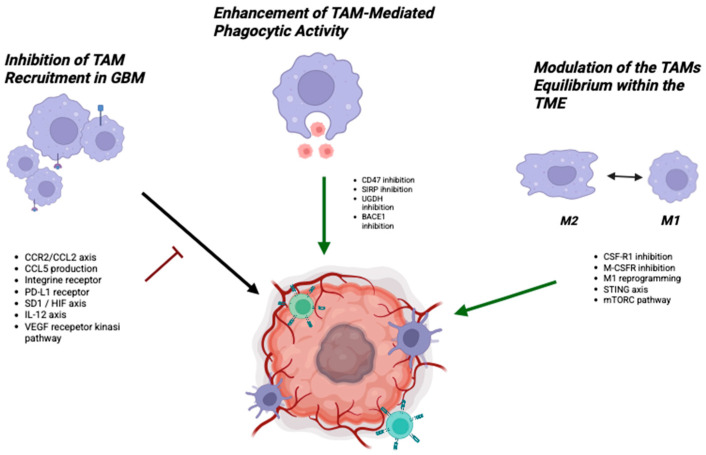
Schematic representation of pharmacological strategies targeting tumor-associated macrophages (TAMs) in glioblastoma (GBM). Red lines indicate inhibitory effects, while green lines indicate stimulatory effects. Therapeutic approaches include the following: (left) inhibition of TAM recruitment (e.g., targeting the CCR2/CCL2 axis, CCL5 production, integrin receptors, PD-L1 receptor, SD1/HIF axis, IL-12 axis, and VEGF receptor kinase pathway); (center) enhancement of TAM-mediated phagocytic activity (via inhibition of CD47, SIRP, UGDH, and BACE1); (right) modulation of the M1/M2 equilibrium (reduced M2 and enhancing M1) within the tumor microenvironment (TME) (through CSF-R1 inhibition, M-CSFR inhibition, M1 reprogramming, STING axis modulation, and mTORC pathway targeting).

**Table 1 cancers-17-02687-t001:** List of studies on pharmacological molecules with an effect on the recruitment of TAM in GBM.

Authors	Study Design	Molecule	Action Mechanism
Flores-Toro et al. [11]	In vivo (murine)	CCX872	CCR2 inhibition
Cho et al. [12]	In vivo (murine)	mNOX-E36	CCL2 inhibition
Zhang et al. [13]	Ex vivo (glioma cells)	Ginsenoside RK3	PPARG downregulation
Tian et al. [14]	In vivo (murine)	OV-Cmab-CCL5	CCL5 production
Miyazaki et al., 2020 [15]	In vivo (murine)	IPI-549	PDL-1 inhibition
Saha et al. [16]	In vivo (murine)	oHSV G47Δ	IL-12 agonism
Deng et al. [17]	In vivo (murine)	OLA-PEG	SDF-1α inhibition
Qiao et al. [18]	Ex vivo (glioma cells)	ZOL@CNPs	HIF-1α inhibition
Saha et al. [19]	In vivo (murine)	Axitinib	VEGFR kinase inhibition
Koula et al. [20]	In vivo (murine)	CeNP + Dox + RGD nanoparticles	Integrin receptors targeting
J. Rubin et al. [21]	In vivo (murine)	AMD 3100	SDF-1α inhibition

**Table 2 cancers-17-02687-t002:** List of studies on pharmacological molecules with an effect on phagocytic activity of TAM in GBM.

Authors	Study Design	Molecule	Action Mechanism
Gholamin et al. [29]	In vitro, in vivo	Hu5F9-G4	SIRPα inhibition
Von Roemeling et al. [32]	In vivo (murine)	Anti-CD47 Ab	CD47 inhibition
Gholamin et al. [33]	In vivo (murine)	Anti-CD47 Ab	CD47 inhibition
Li et al. [34]	Ex vivo (glioma cells)	Nanoparticles loaded with TMZ, anti-CD47 Ab, and PA1094T	Anti-CD47, direct chemo-phototherapy
Xu et al. [35]	In vivo (murine)	oHSV expressing anti-CD47 IgG	CD47 inhibition
Wang et al. [36]	In vivo (murine)	PTX filament hydrogel containing aCD47	CD47 inhibition
Zhai et al. [37]	In vivo (murine)	MK-8931	BACE1 inhibition
Zhan et al. [38]	In vivo (murine)	4-MU	UGDH inhibition

**Table 3 cancers-17-02687-t003:** List of studies on pharmacological molecules with an effect on the modulation of the TAMs equilibrium within the TME.

Authors, Year	Study Design	Molecule	Action Mechanism
Pyonteck et al. [43]	In vivo (murine)	BLZ945	CSF-1R inhibition
Przystal et al. [44]	In vivo, ex vivo	Anti-CSF-1R Ab	CSF-1R inhibition
Van Overmeire et al. [45]	In vivo (murine)	Anti-M-CSFR	M-CSFR inhibition
Butowski et al. [46]	In vivo (phase II trial)	PLX3397	CSF-1R/KIT inhibition
Wang et al. [47]	In vivo (murine)	CCA-M1EVs	M1 reprogramming
Berger et al. [48]	In vivo (murine)	ADU-S100	STING activation
Cong et al. [49]	In vivo (murine)	AZD8055, WYE-125132, MTI-31, rapamicin	mTORC1/2 inhibition
Angelova et al. [50]	In vivo (phase I/IIa trial)	H-1PV	Increased cytotoxic T-cells, activation of TAMs and microglia, etc.
Geletneky et al. [51]	In vivo (murine)	H-1PV	Increased cytotoxic T-cells, activation of TAMs and microglia, etc.
Zheng et al. [52]	In vivo (murine, phase I/IIa trial)	oH2	Direct tumor toxicity, increased anti-tumor immune response
Van Putten et al. [53]	In vivo (phase I trial)	DNX-2401	CDD192, TLR4, and CD64 expression, CD206 reduction

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
