# Peer review of "Targeting Macrophages in Glioblastoma: Current Therapies and Future Directions"

_cancers, 2025, doi:10.3390/cancers17162687_

Round 1
Reviewer 1 Report
Comments and Suggestions for Authors
Overall very well-written.
Not clear whether bone marrow–derived macrophages and myeloid derived suppressor cells (MDSCs) are the same. The paper implies that they are different. Assuming this is the case, please clarify how they differ. MDSCs seem relevant as part of targeting TAMs. Please clarify if this is true. If so, a paragraph summarizing work on targeting MDSCs should be included.
Line 222: Please add reference for "Similarly, anti-VEGF therapies, such as bevacizumab, have been observed to raise the expression of HIF-1α and SDF-1α, contributing to the tumor's ability to evade treatment."
Line 232: Please add reference for "While the increase in macrophages following anti-VEGF therapy is widely accepted,"
This paper would be stronger with a diagram showing the various pathways involved with TAM activation/inhibition that leads to tumor growth. Does not necessarily need to include all the drugs but the 3 major strategies for targeting TAMs would be great if possible.
Small points:
Line 55: bevacizumab and tumor treating fields are also part of standard therapy in GBM. Please add.
Line 303: "inhibition" should be "inhibitor"
Line 435: "valuing" - did you mean "evaluating?"
Author Response
Dear Editor,
Thank you for the opportunity to revise the manuscript. We carefully read the Editor’s and the Reviewer’ comments and we revised the manuscript accordingly.
As follows, we list all the changes that we have made to the manuscript.
Reviewer 1
Not clear whether bone marrow–derived macrophages and myeloid derived suppressor cells (MDSCs) are the same. The paper implies that they are different. Assuming this is the case, please clarify how they differ. MDSCs seem relevant as part of targeting TAMs. Please clarify if this is true. If so, a paragraph summarizing work on targeting MDSCs should be included.
Line 222: Please add reference for "Similarly, anti-VEGF therapies, such as bevacizumab, have been observed to raise the expression of HIF-1α and SDF-1α, contributing to the tumor's ability to evade treatment."
- Response: Thank you for the insightful suggestion. We have clarified the concept as follows: “ While both bone marrow–derived macrophages (BMDMs) and myeloid-derived suppressor cells (MDSCs) originate from the bone marrow, they represent distinct myeloid cell populations with different stages of maturation and immunological roles. BMDMs are fully differentiated macrophages recruited into the glioma microenvironment, often assuming an M2-like phenotype that promotes tumor growth. In contrast, MDSCs are immature myeloid cells with potent immunosuppressive activity that inhibit T cell responses and can differentiate into various cell types, including macrophages. MDSCs, particularly the monocytic subtype (M-MDSCs), are increasingly recognized as relevant contributors to the immunosuppressive milieu in GBM and share pathways with TAM recruitment.
Therapeutic strategies targeting MDSCs, such as CCR2 inhibition, overlap with TAM-directed approaches. For example, the CCR2 antagonist CCX872 reduces both TAM and MDSC infiltration, enhancing the efficacy of checkpoint inhibitors. These findings suggest that targeting MDSCs is a promising and complementary avenue within TAM-modulating strategies”. From line 166 to 177
Line 232: Please add reference for "While the increase in macrophages following anti-VEGF therapy is widely accepted,"
- Response: we added refences 19 (line 246)
This paper would be stronger with a diagram showing the various pathways involved with TAM activation/inhibition that leads to tumor growth. Does not necessarily need to include all the drugs but the 3 major strategies for targeting TAMs would be great if possible.
- Response:we added Figure 2
Small points:
Line 55: bevacizumab and tumor treating fields are also part of standard therapy in GBM. Please add.
- Response: we added in line 55-56.
Line 303: "inhibition" should be "inhibitor"
- Response: We corrected the English form
Line 435: "valuing" - did you mean "evaluating?"
- Response: We have reformulated the phrase.
Reviewer 2 Report
Comments and Suggestions for Authors
The authors set out to summarize the relationship of macrophages in the most malignant brain tumor, glioblastoma. To accomplish this they assembled 30 publications from a literature search from a much larger set of papers. While I have no issues with this approach, I found a number of deficiencies.
First, I felt that while they summarized the papers, they did not really offer much of a synthesis of their findings (other than to note this is a complicated field); there is enough in the literature so that they should be able to objectively evaluate the potential of these strategies.
Second, the manuscript would benefit from one or more summary illustrations demonstrating the potential roles of macrophages in GBM.
Third, while they discuss links with angiogenesis and anti-tumor immunity, their synthesis appears incomplete. For example, there is minimal mention of Plerixafor, wherein there is an ample literature, including clinical findings. They also do not discuss the idea of vasculogenesis versus angiogenesis and leave out the role in these cells in determining radiation response. Finally, they barely touch on the evolving clinical experience with this strategy.
I assume these deficiencies arise from their choice of search strategy and would suggest it be refined. There are many more papers on this topic that have been written and which deserve mention.
Author Response
Reviewer 2
The authors set out to summarize the relationship of macrophages in the most malignant brain tumor, glioblastoma. To accomplish this they assembled 30 publications from a literature search from a much larger set of papers. While I have no issues with this approach, I found a number of deficiencies.
First, I felt that while they summarized the papers, they did not really offer much of a synthesis of their findings (other than to note this is a complicated field); there is enough in the literature so that they should be able to objectively evaluate the potential of these strategies.
- Response: We improved the section Discussion to enhance the synthesis of our findings (from line 477 to 486)
Second, the manuscript would benefit from one or more summary illustrations demonstrating the potential roles of macrophages in GBM.
- Response: We added Figure 2
Third, while they discuss links with angiogenesis and anti-tumor immunity, their synthesis appears incomplete. For example, there is minimal mention of Plerixafor, wherein there is an ample literature, including clinical findings. They also do not discuss the idea of vasculogenesis versus angiogenesis and leave out the role in these cells in determining radiation response.
- Response: paragraph 3.2 (lines 254–268) has been revised for clarity and content, the PRISMA diagram and tables have been updated, and the CXCL12 axis (also referred to as SDF-1α) has been discussed in greater depth, with an expanded description of the associated pathway. Also the role of radiation has been improved from line 263 to 277.
Finally, they barely touch on the evolving clinical experience with this strategy.I assume these deficiencies arise from their choice of search strategy and would suggest it be refined. There are many more papers on this topic that have been written and which deserve mention.
- Response: We thank the reviewer for this insightful comment. As suggested, we have acknowledged this important limitation by introducing a dedicated paragraph in the Limitationssection of the manuscript. In this paragraph, we clarify that our review focused specifically on molecular strategies targeting TAMs in glioblastoma, and we recognize that this focus may have excluded related areas such as the effects of radiotherapy, vasculogenesis, and broader clinical experiences. We agree that the literature in this field is extensive and evolving, and we have now explicitly addressed this limitation to improve the transparency and scope of our synthesis. (line 524 to 534)
Reviewer 3 Report
Comments and Suggestions for Authors
The manuscript by Pennisi et al. provides a comprehensive and thorough overview of the glioblastoma microenvironment, with a particular focus on the characterization of macrophage phenotypes associated with Glioblastoma and the potential therapeutic approaches targeting them. The authors emphasize the importance of macrophage markers in the context of GB and present a substantial amount of valuable information. While the manuscript is generally well written and the discussion of macrophage markers is detailed, some aspects remain unclear—particularly the differences in macrophage profiles before and after treatment. Additionally, the manuscript does not mention whether a Phase I clinical trial with negative outcomes has been conducted, which would be relevant to contextualize the current therapeutic strategies.
Major Revisions
Expand the Introduction:
The introduction is relatively brief compared to other sections of the manuscript. The authors are encouraged to expand this section to provide a stronger contextual foundation for the topic.
Include Classical Macrophage Classification:
The manuscript should include a more detailed description of the classical macrophage classification system. In particular, the authors should discuss the M1/M2 paradigm and elaborate on the alternative macrophage subtypes (M2a, M2b, M2c.) along with their associated markers.
Address Low-Grade Glioma and Microenvironment:
It is recommended that the authors incorporate relevant information on the tumor microenvironment of low-grade gliomas, if available, to provide a more comprehensive perspective.
Incorporate Recent Literature on UDP Release:
The discussion on UDP release and its role in the tumor microenvironment should be strengthened with more recent and relevant literature.
Therapeutic Context and Marker Association:
The manuscript would benefit from a more focused analysis of the relationship between therapeutic interventions and associated molecular markers. Including a summary table could enhance clarity and reader comprehension.
Improve Data Presentation in Tables:
The quality and clarity of the data presented in the tables need to be improved. Consistency in formatting, labeling, and units should be ensured for better readability.
Add a Schematic Overview:
For improved understanding, the authors should consider including a schematic diagram that illustrates the glioblastoma microenvironment and outlines potential therapeutic strategies.
Minor Revision
Add Suggested Citation:
The following reference should be included and cited appropriately in the manuscript:
Science. 2016 May 20;352(6288):aad3018. doi: 10.1126/science.aad3018.
Author Response
Reviewer 3
The manuscript by Pennisi et al. provides a comprehensive and thorough overview of the glioblastoma microenvironment, with a particular focus on the characterization of macrophage phenotypes associated with Glioblastoma and the potential therapeutic approaches targeting them. The authors emphasize the importance of macrophage markers in the context of GB and present a substantial amount of valuable information. While the manuscript is generally well written and the discussion of macrophage markers is detailed, some aspects remain unclear—particularly the differences in macrophage profiles before and after treatment. Additionally, the manuscript does not mention whether a Phase I clinical trial with negative outcomes has been conducted, which would be relevant to contextualize the current therapeutic strategies.
- Response: We sincerely thank you for your constructive and thoughtful review of our manuscript. We greatly appreciate your positive comments regarding the comprehensiveness, scientific depth, and relevance of our work. We have carefully addressed all your suggestions and revised the manuscript accordingly to improve its clarity, depth, and scientific value. We would like to clarify that our manuscript already includes references to both Phase I and Phase II clinical trials relevant to TAM-targeting strategies in glioblastoma. Specifically:
At lines 401, we discuss the Phase I/IIa ParvOryx01 clinical trial evaluating the oncolytic protoparvovirus H-1PV, which demonstrated safety and the ability to cross the blood–brain barrier, as well as immunomodulatory effects such as increased infiltration of cytotoxic T cells and TAM activation (Angelova et al. [44], Geletneky et al. [45]). At lines 408, we describe clinical evaluation of DNX-2401, an oncolytic adenovirus, as part of a Phase I trial, which showed good tolerability and signs of immune activation (Van Putten et al. [47]).
At lines 451, we report the Phase II trial of PLX3397, a CSF1R/KIT inhibitor, which—despite favorable pharmacokinetics and tolerability—failed to show significant clinical benefit in GBM patients (Butowski et al. [40]).
Expand the Introduction:
The introduction is relatively brief compared to other sections of the manuscript. The authors are encouraged to expand this section to provide a stronger contextual foundation for the topic. Classical Macrophage Classification:
The manuscript should include a more detailed description of the classical macrophage classification system. In particular, the authors should discuss the M1/M2 paradigm and elaborate on the alternative macrophage subtypes (M2a, M2b, M2c.) along with their associated markers.
Address Low-Grade Glioma and Microenvironment:
It is recommended that the authors incorporate relevant information on the tumor microenvironment of low-grade gliomas, if available, to provide a more comprehensive perspective.
- Response: We thank the reviewer for these valuable suggestions. In response, we have substantially expanded the Introduction section (lines 68–86) to provide a stronger conceptual foundation for the manuscript. The revised version includes: A more detailed explanation of the classical macrophage classification system, including the M1/M2 paradigmand the description of alternative M2 subtypes (M2a, M2b, M2c) and their associated stimuli and functional markers. Although the manuscript already included a dedicated section on macrophage phenotypes (Section 3.1), we agree that briefly introducing these concepts in the Introduction improves readability and contextual flow. A concise paragraph addressing the tumor microenvironment of low-grade gliomas (LGGs), highlighting differences in immune infiltration and TAM polarization compared to GBM. While LGGs are not the primary focus of this review, we acknowledge that providing this comparison offers a broader and more informative perspective for the reader.
Incorporate Recent Literature on UDP Release:
The discussion on UDP release and its role in the tumor microenvironment should be strengthened with more recent and relevant literature.
Therapeutic Context and Marker Association:
The manuscript would benefit from a more focused analysis of the relationship between therapeutic interventions and associated molecular markers. Including a summary table could enhance clarity and reader comprehension.
Improve Data Presentation in Tables:
The quality and clarity of the data presented in the tables need to be improved. Consistency in formatting, labeling, and units should be ensured for better readability.Add a Schematic Overview:
For improved understanding, the authors should consider including a schematic diagram that illustrates the glioblastoma microenvironment and outlines potential therapeutic strategies.
- Response: Thank you for these insightful recommendations. We have significantly enhanced the UDP-related discussion by integrating the latest evidence on how extracellular UDP can function as a DAMP via activation of the P2Y₆ receptor, influencing TAM polarization and immune activity. In particular, we now describe mechanistic insights from recent studies demonstrating that inhibition of the UDP–P2Y₆ pathway can shift TAMs toward a pro-inflammatory phenotype, promote cytotoxic T-cell infiltration, and improve responsiveness to checkpoint blockade. These updates are incorporated into the UDP section (lines 229- 359). Additionally, we have refined the Therapeutic Context and Marker Association section by more clearly linking specific TAM-targeting strategies to their molecular markers (e.g., CD163, CD206, CD86). To support comprehension, we added a new summary Table 4 explicitly correlating therapies, targets, marker responses, and outcomes, and introduced a schematic Figure 2 that visually maps the tumor microenvironment, TAM origins, and targeted interventions. These improvements substantially bolster the clarity, translational relevance, and visual presentation of the manuscript.
Add a Schematic Overview:
For improved understanding, the authors should consider including a schematic diagram that illustrates the glioblastoma microenvironment and outlines potential therapeutic strategies.
Minor Revision
Add Suggested Citation:
The following reference should be included and cited appropriately in the manuscript:
Science. 2016 May 20;352(6288):aad3018. doi: 10.1126/science.aad3018.
- Response: We thank the reviewer for these constructive suggestions. In response, we have included a new schematic illustration (Figure 2) that provides a visual summary of the glioblastoma microenvironment, highlighting the dual origin of TAMs, their polarization states, and associated therapeutic targets. This figure is intended to enhance reader comprehension and complement the text. Additionally, we have incorporated the suggested citation (Quail & Joyce, Science, 2016) [43]